# Improvement of PBAT Processability and Mechanical Performance by Blending with Pine Resin Derivatives for Injection Moulding Rigid Packaging with Enhanced Hydrophobicity

**DOI:** 10.3390/polym12122891

**Published:** 2020-12-02

**Authors:** Cristina Pavon, Miguel Aldas, Harrison de la Rosa-Ramírez, Juan López-Martínez, Marina P. Arrieta

**Affiliations:** 1Instituto de Tecnología de Materiales, Universitat Politècnica de València (UPV), 03801 Alcoy-Alicante, Spain; miguel.aldas@epn.edu.ec (M.A.); hardela@epsa.upv.es (H.d.l.R.-R.); jlopezm@mcm.upv.es (J.L.-M.); 2Departamento de Ciencia de Alimentos y Biotecnología, Facultad de Ingeniería Química y Agroindustria, Escuela Politécnica Nacional (EPN), Quito 170517, Ecuador; 3Departamento de Ingeniería Química Industrial y del Medio Ambiente, Escuela Politécnica Superior de Ingenieros Industriales, Universidad Politécnica de Madrid (ETSII-UPM), Calle José Gutiérrez Abascal 2, 28006 Madrid, Spain; 4Grupo de Investigación: Polímeros, Caracterización y Aplicaciones (POLCA), 28006 Madrid, Spain

**Keywords:** polybutylene adipate-co-terephthalate (PBAT), pine resin, gum rosin, blends, compatibility effect, plasticizing effect, hydrophobicity, packaging

## Abstract

Polybutylene adipate-co-terephthalate (PBAT) is a biodegradable polymer with good features for packaging applications. However, the mechanical performance and high prices of PBAT limit its current usage at the commercial level. To improve the properties and reduce the cost of PBAT, pine resin derivatives, gum rosin (GR) and pentaerythritol ester of GR (UT), were proposed as sustainable additives. For this purpose, PBAT was blended with 5, 10, and 15 wt.% of additives by melt-extrusion followed by injection moulding. The overall performance of the formulations was assessed by tensile test, microstructural, thermal, and dynamic mechanical thermal analysis. The results showed that although good miscibility of both resins with PBAT matrix was achieved, GR in 10 wt.% showed better interfacial adhesion with the PBAT matrix than UT. The thermal characterization suggested that GR and UT reduce PBAT melting enthalpy and enhance its thermal stability, improving PBAT processability. A 10 wt.% of GR significantly increased the tensile properties of PBAT, while a 15 wt.% of UT maintained PBAT tensile performance. The obtained materials showed higher hydrophobicity than neat PBAT. Thus, GR and UT demonstrated that they are advantageous additives for PBAT–resin compounding for rigid food packaging which are easy to process and adequate for industrial scalability. At the same time, they enhance its mechanical and hydrophobic performance.

## 1. Introduction

A mismanagement of the disposal of short term polymers after use joined with the impossibility to handle the problem only by mechanical recycling due to still inefficient waste management programs, the composition of some plastic formulations (i.e., blends, composites, nanocomposites, etc.), and the fact that plastics cannot be recycled forever [1], as well as the resistance to degradation by many plastic materials, have led to plastic hoarding in the environment [2,3,4,5]. In this context, the production of biodegradable polymers has considerably increased during recent years, particularly for short term applications, such as food packaging materials [5,6], while resulting in a necessary alternative to deal with the environmental problem produced by the accumulation of plastics in the environment. Naturally occurring microorganisms offer the opportunity to enzymatically degrade biodegradable polymers into small molecules (carbon dioxide and water) [7]. However, biodegradable polymers present reduced overall performance with respect to traditional petroleum-based counterparts, such as higher sensitivity to humidity and thermal degradation, as well as poor barrier and mechanical performance, which hinder its massive industrial exploitation [8,9,10]. Among biodegradable polymers, biopolyesters are positioned in the packaging sector as the most suitable polymers to replace petrol-based plastics in food packaging applications; thus, there are many research studies focused on improving biopolyesters’ performance, including poly(lactic acid) (PLA), polyhydroxyalkanoates (PHAs) and its derivatives [11], poli(ε-carpolactone) (PCL), and poly(butylene adipate-co-terephthalate) (PBAT) [12,13,14]. Aliphatic biopolyesters, such as PLA and PHB, have been, to date, the most promising biodegradable polymers for biodegradable or compostable food packaging products [7]. However, aliphatic biopolyesters present some drawbacks for food packaging applications with respect to their petrochemical counterparts such as sensitiveness to hydrolytic degradation, which is highly influenced by ambient moisture and temperature, leading to low thermal stability [7,15,16]. Additionally, for food packaging purposes, their poor barrier properties and inherent unfavourable physical and mechanical properties, such as their limited stretchability, have limited their commercial utility [7,17]. Thus, many research efforts have been focused on biopolyesters modification for extending their industrial application as flexible materials with improved barrier and hydrophobicity performance, such as blending, the addition of fillers and/or nanofillers, surface plasma treatment, or copolymerization [16,18,19].

To improve the physicochemical properties of aliphatic polyesters without losing their biodegradability, an effective approach that has gained interest is the incorporation of aromatic ester moieties in the polyester chain [20,21]. Taking this into consideration, co-polyesters have been synthesized by melt polycondensation and melt transesterification reactions of poly(butylene adipate) (PBA) and poly(butylene terephthalate) (PBT) under the generic name of polybutylene adipate-co-terephthalate (PBAT) [22,23]. PBAT is a biodegradable synthetic aliphatic–aromatic thermoplastic co-polyester with low crystallinity [24,25]. PBAT have similar thermal and mechanical properties to some polyethylenes and present more flexibility than other biodegradable biopolyesters (i.e., PLA and PHB), which make it suitable for packaging applications [22]. Moreover, PBAT exhibits higher hydrophobicity and is easy to process [26]. Nevertheless, the mechanical properties of PBAT are often insufficient for various end-use applications, and the prices of PBAT are high [27]. For instance, injection moulding is one of the most used polymer processing technologies for rigid packaging manufacturing at the industrial level. Polymers for injection moulded rigid packaging are required to have high mechanical performance to overcome the strong shear stresses during the injection moulding process and during service [28].

Thus, as was already commented, there are many strategies to improve biopolymer performance for extending its industrial applications in the food packaging sector. In this sense, blending strategies allow tuning of the physical and mechanical properties of biopolymers by a relatively simple approach with potential scalability to the industrial sector [18]. Thus, blending PBAT with low cost polymers or additives is an attractive alternative to improve its overall performance and exploit the good features of PBAT [24].

On the other hand, there are many abundant, non-toxic, inexpensive, and eco-friendly materials derived from biomass that are attractive candidates as polymer additives [26]. In this context, gum rosin, a by-product of pine resin, has gained attention as a monomer for polymer synthesis [29] and as an additive in the plastic processing industry [30,31]. The structure of gum rosin can be modified due to its acidity and hydrophobicity [32,33,34]. For instance, pentaerythritol alcohol is used to stabilize gum rosin to increase its thermal resistance [30,35]. Gum rosin and its derivatives are fragile and rigid due to their hydrogenated phenanthrene ring structure [36,37]. Therefore, they have been used as environmental friendly additives in both thermoplastics and thermosets [30,38,39,40]. Nevertheless, the use of gum rosin and its derivatives as additives for biopolymeric matrices is relatively new. Consequently, the literature in the field is limited. Recently, Aldas et al. (2019) have studied the effect of gum rosin and gum rosin esters in a commercial thermoplastic starch-based polymeric matrix based on Mater-Bi^®^-type bioplastic and found that gum rosin plasticized the polymeric matrix and that rosin esters acted as compatibilizers, enhancing the toughness of the Mater-Bi^®^ polymeric matrix [30,35]. Pavon et al. (2020) studied the addition of raw gum rosin in PCL and found that gum rosin plasticizes the PCL matrix and improved its thermal stability [37]. Aldas et al. (2020) analysed the effect of different gum rosin derivatives (gum rosin modified with maleic anhydride, disproportionated gum rosin, and two gum rosin esters) on a pure TPS matrix and concluded that the use of gum rosin and gum rosin derivatives not only stiffened the TPS polymeric matrix, but also ensure the thermal stability in the extrusion and injection moulding processes [28]. Moreover, De la Rosa el al. (2020) added gum rosin to PLA matrix and established that GR and a rosin ester provide lubricating effects over the PLA polymeric chains, enhancing the processability of the formulations. Finally, it is worth mentioning that Moustafa et al. (2017) studied PBAT-PLA blends loaded with organoclay modified with gum rosin for green packaging purposes. They found that gum rosin improves the compatibility between PLA and PBAT polymeric matrices in the blend, with a consequent improvement in the viscoelastic and tensile properties. Finally, it was found that gum rosin provided interesting features to the PLA/PBAT matrix for food packaging applications such as antimicrobial activity [41]. Nevertheless, to the best of our knowledge, this is the first time that gum rosin and pentaerythritol ester of GR have been used to develop PBAT–gum rosin-based injection moulded materials.

The present study aims to explore for the first time the use of pine resin derivatives as additives to improve the processability as well as to enhance the toughness and water resistance performance of PBAT. For this purpose, gum rosin and a pentaerythritol ester of gum rosin were blended with PBAT matrix in three different amounts: 5, 10, and 15 wt.%. The blends were melt-extruded and further processed by injection moulding to simulate the industrial processing conditions for rigid packaging production. The influence of gum rosin and the pentaerythritol gum rosin ester on the processing as well as on the mechanical, thermal, and microstructural properties was studied. Finally, the surface wettability was studied with the aim to evaluate the potential applications of this materials in humid conditions.

## 2. Materials and Methods

### 2.1. Materials

Polybutylene adipate-co-terephthalate (PBAT) BiocosafeTM 2003 F was kindly supplied by Xinfu Pharmaceutical Co. Ltd. (Zhejiang, China). The commercial grade is characterized by a density of 1.25 g/cm^3^ and a melt flow index < 6 g/10 min at 190 °C. Two pine resin derivatives were used as additives—gum rosin (label as GR, softening point of 76 °C, acid number 167) supplied by Sigma-Aldrich (Mostoles, Spain) and Unik Tack P100 resin, a pentaerythritol ester of gum rosin, (label as UT, softening point of 90 °C, acid number 15), kindly supplied by United Resins (Figueira da Foz, Portugal).

### 2.2. Methods

#### 2.2.1. Miscibility Prediction

The solubility parameters of pine resins were calculated to predict the compatibility of PBAT with pine resin derivatives using Equation (1):(1)$$\delta =\frac{D\sum G}{M}$$
where *δ* is the solubility parameter ((cal cm^−3^)^1/2^ mol^−1^), *D* is the density (g cm^−3^), *G* is the group molar cohesive energy ((cal cm^−3^)^1/2^ mol^−1^), and *M* is the molar mass per repetitive unit (g mol^−1^).

#### 2.2.2. PBAT-Resin Formulations Preparation

The PBAT resin-based formulations were prepared mixing 5, 10, and 15 wt.% of pine resin derivatives (GR and UT) in the PBAT polymeric matrix. Six formulations were obtained. Before the mixing, the materials were conditioned as follows: PBAT was dried overnight at 40 °C [42], while GR and UT were dried at 50 °C for 24 h in an air circulation oven [30]. First, the formulations were processed in a twin-screw extruder (Dupra S.L., Castalla, Spain), with a temperature profile of 180, 170, 160, and 150 °C (from die to hopper) at 50 rpm. Later, the obtained PBAT resin-based materials were pelletised [30], and further injection moulded into test specimens in an injection moulding machine Sprinter-11, Erinca S.L. (Barcelona, Spain), with a temperature profile of 180, 170, 160, 150, and 140 °C, from die to hopper, as is schematically represented in Scheme 1. The chemical structure of PBAT and both pine resin derivatives used here are also shown in Scheme 1. Prior to characterization, injection moulded samples were conditioned at 25 °C and 50 ± 5% RH for 24 h [4].

#### 2.2.3. Mechanical Characterization

The tensile properties of PBAT resin-based formulations were assessed in a universal test machine (Ibertest Elib 30 of SAE Ibertest (Madrid, Spain)) at room temperature, according to ISO 527-1 [43]. The tests were carried out with a load cell of 5 kN and a crosshead rate of 500 mm/min. Specimens in a dog bone-shape “1BA” type (80 × 10 × 4 mm) according to ISO 527-2 [44] were used. The determined parameters were Young’s modulus (MPa), elongation at break (%), and tensile strength (MPa). Young’s moduli were calculated from the initial slope of the stress–strain curves (0.05–0.25% strain range). The hardness test was performed in the Shore D durometer 673-D by Instruments J. Bot S.A. (Barcelona, Spain) in rectangular-shape specimens with dimensions: 80 ± 2.0 × 10 ± 0.2 mm^2^ and 4 ± 0.2 mm thickness. Five specimens for each formulation were tested out and the mean and standard deviation of each mechanical property are reported.

#### 2.2.4. Microstructural Characterization

Field emission scanning electron microscopy (FESEM) was conducted on a Zeiss Ultra 55 microscope at 1 kV over the cryofracture surface of the samples. Then, the samples were mounted on aluminium stubs using double sided adhesive tape and coated with a gold-palladium alloy layer for improved conduction. The samples were kept in a vacuum chamber prior to analysis.

#### 2.2.5. Thermal Characterization

Differential scanning calorimetry (DSC) tests and thermogravimetric analyses (TGA) were used to determine the thermal parameters of PBAT and PBAT resin-based formulations. For the DSC analyses, samples (5–10 mg) were tested, under nitrogen flow of 30 mL/min at a heating rate of 20 °C/min, in a differential scanning calorimeter Mettler Toledo 821 (Mettler Toledo, Schwerzenbach, Switzerland). A three-stage temperature program was used: heating from 25 to 260 °C; cooling to −50 °C; and reheating to 300 °C. Melting temperature and crystallization temperature were recorded. 

The TGA analysis was carried out in a thermogravimetric analyser TGA PT1000 from Linseis (Selb, Germany). Samples (15–20 mg) were scanned from 35 to 700 °C at a heating rate of 10 °C/min nitrogen atmosphere with a flow rate of 30 mL/min. The initial (T5%) and final (T90%) degradation temperature were reported as the temperature to which the material loses 5% and 90% of its initial mass, respectively. Additionally, the maximum degradation rate temperature (Tmax) was determined at the peak of the first derivative of the TGA curve (DTG).

#### 2.2.6. Dynamic Mechanical Thermal Characterization

Dynamic mechanical thermal analysis (DMTA) was carried out in a DMA1 Mettler-Toledo (Schwerzenbach, Switzerland) in a single cantilever mode. Samples of 20 ± 2.0 × 4.5 ± 0.2 mm^2^ with an average thickness of 1 ± 0.2 mm were subject to a temperature sweep from −100 up to 80 °C at a heating rate of 2 °C/min, at a frequency of 1 Hz with a maximum shear deformation (%γ) of 0.1%. The dynamic storage modulus (E′) and loss factor (tan (δ)) were measured as a function of temperature. The main relaxation temperature (Tα), associated with the glass transition temperature of the samples, was determined as the temperature at the maximum of the tan (δ) peak displayed in the tan (δ) versus temperature curves.

#### 2.2.7. Wettability

Surface wettability was determined through water contact angles (WCA) measurements of deionized water on the surface of the PBAT resin-based specimens by the sessile drop method. At least six water droplets (≈1.5 µL) were randomly deposited on the sample surface with a precision syringe at room temperature and the water contact angle was measured eight times for each droplet at room temperature using an optical goniometer EasyDrop-FM140 from Kruss Equipments (Hamburg, Germany).

#### 2.2.8. Colour Characterization

The surface colour of each formulation was measured with a Colorflex-Diff2 458/08 colorimeter from HunterLab (Reston, VA, USA). The test was performed under the CIE L*a*b* colour space and 10 measurements on each sample were performed. The CIE L*a*b* is a 3 dimensional model, where L* represents the lightness and ranges from 0 (pure black) to 100 (diffuse white), the chromatic a* axis extends from green (−a*) to red (+a*), and the chromatic b* axis extends from blue (−b*) to yellow (+b*) [8,45]. L*, a*, and b* coordinates were reported along with the yellowness index (YI). Furthermore, the total colour difference (∆E) was obtained as a numerical comparison of each formulation colour with the colour of the standard (taken PBAT as standard) and was calculated using Equation (2)
(2)$$∆E=\sqrt{∆{a*}^{2}+∆{b*}^{2}+∆{L*}^{2}}$$

#### 2.2.9. Statistical Analyses

All the data were statistically analysed in OriginPro 8 software from OriginLab (Northampton, MA, USA). The significant differences were assessed at 95% confidence level according to Tukey’s test using a one-way analysis of variance (ANOVA). A comparison for all pairwise differences between factor level means is shown in each table and/or figure.

## 3. Results

### 3.1. Miscibility

It is known that the more similar the chemical structure of the additives and polymeric matrix, the higher the potential is that one is dissolved in the other. A widely used method to predict the thermodynamic miscibility of polymer blends is by means of the calculation of the solubility parameters. The solubility parameter could be evaluated based on the chemical structure of the polymer using group contribution models. In fact, polymers can be well dissolved in plasticizers and/or additives when their solubility parameters are similar. The solubility parameter for PBAT ranges between 20.5 [46] and 22.2 MPa^1/2^ [47], while the solubility parameter of each part of the co-polymer has been reported as 21.6 and 17.9 MPa^1/2^ for PBT and PBA, respectively [46]. The solubility parameters for GR and UT were calculated as 10.0 and 11.9 MPa^1/2^, respectively. The solubility parameters are different, while there are less differences with the aliphatic segment of the PBAT and consequently, better miscibility between PBAT and resins should be expected with this segment of the co-polymer.

### 3.2. Mechanical Characterization

Figure 1 shows the effect of increasing the amount of pine resin derivatives on the tensile properties of PBAT resin-based formulations. The results show that neat PBAT is a rubbery polymer with a tensile strength of 12.19 ± 0.62 MPa, Young’s modulus of 74.19 ± 1.55 MPa, and a superior elongation at break of 516.47 ± 13.20%. In Figure 1a, the tensile strength of all the formulated materials is presented. When GR was added to PBAT in a 5 wt.%, no significant differences were detected (*p* > 0.05) in the tensile strength. On the contrary, 10 wt.% of GR in PBAT significantly increases the tensile strength (*p* < 0.05) by 27%, showing a compatibilizing effect of GR at this composition. However, when GR composition was 15 wt.%, an abrupt drop in tensile strength was recorded, 28% lower than neat PBAT. This indicates that at 10 wt.% of GR, the maximum values of the tensile strength are achieved. On the other hand, UT did not significantly change the tensile strength of PBAT when added in 5, 10, or 15 wt.%, contrary to other polymeric matrices such as PLA where the pentaerythritol ester decreases it [48] or thermoplastic starch (TPS) where the maximum of mechanical properties was found at 5 wt.% of UT, as stated by Aldas et al. (2019) [30]. A similar behaviour in the tensile strength due to GR addition was found by de la Rosa-Ramíez et al. (2020), who blended PLA with GR and determined that the tensile strength decreases in compositions above 5 wt.% [48].

A comparison among Young’s moduli of the materials is shown in Figure 1b. It is noticeable that a 5 wt.% of either GR or UT significantly reduces the PBAT’s Young’s modulus more than 30% (*p* < 0.05). Even more, 10 wt.% of GR significantly increased the PBAT’s Young’s modulus, reaching the best results for GR; therefore, by adding 15 wt.%, Young’s modulus decreases to a value statistically equal to PBAT-5GR. In contrast, an increment of UT content to 10 and 15 wt.% produces a gradual increase in Young’s modulus, still showing miscibility. However, Young’s moduli of PBAT-UT formulations are not higher than the modulus of neat PBAT. The reduction in Young’s modulus in PBAT-15GR along with the reduction in tensile strength in the same formulation suggest a plasticization effect provided by the addition of GR to PBAT. This plasticization effect was also reported when GR was used in 10 wt.% in PCL [37], in 15 wt.% in thermoplastic starch-based materials (Mater-Bi^®^ NF 866) [49], and in different proportions in PLA [31].

The elongation at break results (Figure 1c) reveals that low amounts of both GR and UT resins produce a plasticization effect, in good accordance with the reduction in Tg_2_ (Table 1) of the hard segments of PBAT, as will be further discussed in DMTA section. Figure 1c corroborates that the highest amount of GR to obtain a miscible blend is 10 wt.% content; the elongation at break of PBAT-GR formulations has its maximum value 14% higher than neat PBAT. GR caused a significant increase in the elongation at break (when blended in 5 and 10 wt.%) or stays statistically equal (at 15 wt.%) than PBAT. Furthermore, it is seen that UT preserves PBAT elongation at break with no statistically significant changes (*p* > 0.05) for contents of 5, 10, and 15 wt.%.

Regarding the hardness (Figure 1d), it could be observed that both GR and UT significantly increased the hardness of PBAT when added in 5 and 10 wt.% (*p* < 0.05). Moreover, no significant differences were observed between the hardness of these materials in the mentioned resin contents. GR in 15 wt.% produces no significant change in the hardness of the formulation with respect to neat PBAT (*p* > 0.05). The behaviour of PBAT-GR hardness verifies that the highest amount of the PBAT-GR blend to obtain miscible formulations is 10 wt.% of GR. A similar effect in this parameter was observed by Aldas et al. (2019) [30]. Meanwhile, the hardness of UT at 5, 10, and 15 wt.% presents no statistical differences.

The tensile test results prove that 10 wt.% of GR significantly increased the tensile strength and toughness of PBAT. Meanwhile, 15 wt.% UT preserved PBAT tensile properties. Additionally, UT significantly increases the hardness of PBAT in all the contents, while GR does so in 5 and 10 wt.% and keeps the PBAT value at 15 wt.%. Accordingly, the use of pine resin derivatives is favourable, not only as sustainable materials, but also as additives that can enhance PBAT tensile and hardness properties or maintain them while keeping or reducing the production costs of the final material.

### 3.3. Microstructural Characterization

The microstructures of the cryo-fractured surface of PBAT and PBAT-resin-based formulations are shown in Figure 2. The surface morphology of neat PBAT (Figure 2a) shows a typical rigid cryofracture surface with a rough structure [50,51]. In general, no phase separation is observed in the formulated injected moulded materials.

A content of 5 wt.% of GR smoothes the fracture structure (Figure 2b), which reveals GR compatibility with PBAT, in accordance with the increase in the ductility and in the elongation at break in the material. In PBAT-10GR (Figure 2d), the compatibility is enhanced, as the PBAT surface is completely flat and smooth. This compatibility between PBAT and GR agrees with the increase in the toughness of the PBAT-10GR blend. The compatibility of PBAT with GR was also studied by Moustafa et al. (2017), who have found a compatibilization between PBAT and PLA polymeric matrices with the addition of GR [41]. Meanwhile, PBAT-15GR shows a rough surface with small fractions of GR that seem to not be completely solubilized in the polymeric matrix (see red arrows in Figure 2f). This suggests reduced compatibility in the interfacial adhesion between PBAT and GR, and explains the low mechanical properties obtained for this formulation.

Regarding the rosin ester (UT), it is seen that 5 wt.% of UT produces a marginal change in the fractured surface of PBAT (Figure 2c). Though, as the content of UT increased, the structure became smother than that of neat PBAT (Figure 2c,e,g), which explains the decrease in Young’s modulus in 5 and 10 wt.% of UT. In 15 wt.% UT, the surface is more smooth, pointing to a better compatibilization between PBAT and UT than in the other PBAT-UT-based formulations. However, none of the PBAT-UT blends exhibit a flat surface as PBAT-10GR does, which suggest that the UT is miscible with PBAT in the whole range of the studied contents. Therefore, the best mechanical properties among the studied formulations are reached with 10 wt.% of GR in the PBAT matrix (PBAT-10GR). The materials that contain UT, in all the studied compositions, do not reach the mechanical performance of PBAT-10GR.

### 3.4. Thermal Characterization

The melting and crystallization behaviour of PBAT and the PBAT-resin-based formulation were determined throughout the DSC curves (Figure 3). Additionally, the thermal data of all the formulations are summarized in Table 1. It is seen that the addition of GR in 10 and 15 wt.% reduced the melting temperature of PBAT in 3 and 5 °C, respectively. Meanwhile, 5 wt.% of GR produced no differences in the T_m_ of neat PBAT. UT marginally reduced the melting temperature (1 °C) when added in 5, 10, and 15 wt.%. The reduction in the melting temperature confirms the compatibilizing and plasticizing effect of GR on the PBAT matrix at 10 and 15 wt.%, respectively. Similar findings have already been detected by Aldas et al. (2020) in thermoplastic starch-based materials (Mater-Bi^®^ NF 866) [49]. This behaviour was confirmed by the mechanical properties and FESEM analyses.

Additionally, it is seen that the melting enthalpy of PBAT is reduced, in more than three times its value, when pine resin derivatives were added, except for PBAT-15GR. This result implies that pine resin derivatives at 5 and 10 wt.% improve the processability of PBAT not only because they allow increased polymer chain mobility, but also since less energy is required to melt the blend. It should be highlighted that at an industrial level, one of the focuses of the process to increase the energy efficiency for injection moulding is in the melting of the polymer [52]. Moreover, in PBAT-15GR, the formation of different PBAT crystal structures is promoted as two melting peaks are observed [53]. This behaviour can be ascribed to the fact that GR is not effectively solubilized in the PBAT matrix at this proportion and facilitates the crystallization of PBAT low melting components during processing. This result is in good agreement with the mechanical characterization (Figure 1). Furthermore, a decrement in the crystallization temperature, between 8 and 13 °C, is seen in all the formulations with GR and UT. A decrease in crystallization temperatures suggests that pine resin derivatives increase PBAT chain mobility, which is attributed to a plasticizing effect of these additives [54]. The same effects on the crystallization temperature with the addition of pine resin derivatives were reported by Pavon et al. (2020) using GR as additive for the PCL polymeric matrix [37].

Regarding the thermal stability, the TGA and its first derivative (DTG) curves are depicted in Appendix A. In the case of PBAT, the TGA analysis shows that the thermal degradation occurred in four steps with two main degradation stages. The first one is between 326 and 450 °C and is attributed to the maximum decomposition of aliphatic co-polyester adipic acid and 1,4-butanediol. The second one occurs at around 490 °C, and is associated with the decomposition of aromatic co-polyester terephthalic acid [55,56].

Accordingly, the DTG reveals that the polymer has two maximum degradation temperatures (T_max1_ and T_max2_) at 408.8 and 530.8 °C. Finally, in the fourth step at 580 °C, small weight fluctuations with a tendency to become constant were detected [57]. The TGA curves of PBAT-resin-based formulations also present four steps of degradations. The onset degradation temperature is increased in PBAT-10GR and all PBAT-UT blends in at least 10 °C. However, PBAT-15GR has a lower T_5%_ than PBAT, which may be due to the incomplete solubilization of gum rosin in the PBAT matrix that is less thermally stable [30]. The higher thermal stability provided by UT with respect to GR can be related to the closer solubility parameter of UT with PBAT than GR with PBAT, particularly with the aliphatic part. As well, T_max1_ and T_max2_ are increased in the PBAT-10GR and PBAT-UT blends. This behaviour implies an enhanced thermal stability of the PBAT with the presence of pine resins derivatives (GR and UT).

### 3.5. Dynamic Mechanical Thermal Characterization

Figure 4a exhibits the logarithm of storage moduli (G′) of PBAT and PBAT resin-based formulations with temperature. It is seen that the storage moduli decrease with the temperature increase. In general terms, all the curves present two plateaus in the temperature ranges from −100 to −25 °C and 15 to 40 °C. These plateaus are attributed to the rubbery state of the polybutylene adipate and terephthalate segments of PBAT, respectively [58]. Moreover, two abrupt drops of G′ are seen beyond the plateaus, from 1000 to 100 MPa and from 100 to 10 MPa, which are associated with primary and secondary transitions in the material around −20 and 50 °C, respectively. These transitions correspond to peaks in the tan (δ) curve (Figure 4b). A primary peak is related to T_g1_ of the soft aliphatic segment and a small broad secondary peak corresponds to T_g2_ of the rigid aromatic domain [58,59,60].

The storage modulus (G′) of the PBAT resin-based samples is higher than the PBAT, except for PBAT-15GR, which exhibits an incomplete solubilization of GR into the PBAT matrix already discussed, probably due to an efficient stress transfer from the additive to the matrix [61]. However, for temperatures below −26 °C, the increase in G′ of the formulations with respect to G′ of the PBAT is marginal. Above −26 °C, the differences become noticeable. This is because −26 °C is the glass transition temperature of PBAT. Though, working over T_g_ allows a better motion of the chains which grants the incorporation of the additives in all the structure. Thus, over −26 °C, the effect of stress transfer is upgraded. PBAT-15GR presents a lower storage modulus in all the studied ranges because of the interfacial adhesion problems in the matrix, as seen in the SEM micrographs (Figure 2f).

With respect to the loss factor, plotted in Figure 4b, the incorporation of pine resin derivatives in the PBAT matrix results in increments in the T_g1_ (located at approximately −20 °C) of the aliphatic segment of PBAT (above −26 °C). T_g1_ and T_g2_ of PBAT and PBAT-resin-based formulations are listed in Table 1. The addition of 5 wt.% of GR produced an increment in the T_g1_ of 3 °C with respect to the PBAT matrix, while 5 wt.% of UT produced an increment of 4 °C. PBAT-10GR increased the temperature by 6 °C, while PBAT-10UT increased it by around 4 °C. At a 15 wt.% of GR, the increment of T_g1_ is 9 °C. In contrast, at 15 wt.% of UT, the increment of T_g1_ is only 2 °C. The reported increments in the glass transition temperature reveal an enhancement of the interfacial adhesion between the additive and the polymeric matrix [61]. Notably, GR has a better compatibilization than UT with the aliphatic segment of PBAT matrix, mainly because gum rosin has a solubilizing effect in the PBAT, as stated by Aldas et al. (2020) [49]. However, the shift of T_g1_ of PBAT-15GR could be better related to a segmental immobilization of the matrix chains due to the presence of the additive [57,58], which is in accordance with its low mechanical properties.

The secondary glass transition peak T_g2_, found at around 60 °C and associated with the aromatic rings of PBAT that constitute the hard segments present a higher area in both PBAT-5GR and PBAT-5UT, presented a sharp peak. At the same time, PBAT-10GR shows a broader peak in this region with lower values of tan (δ). Moreover, PBAT-10UT and PBAT-15UT present no differences with respect to neat PBAT. These variations in the curves expose that the compatibility of PBAT-10GR with the hard segment of PBAT is much better that the one of PBAT-5GR and PBAT-5UT [58]. This is because the broadening of the transition region indicates the inhibition of the relaxation process of PBAT [57]. Therefore, the mechanical properties are high and stable in these materials, as seen in the mechanical characterization results (Figure 1). On the other hand, PBAT-10UT and PBAT-15UT have a more similar behaviour than PBAT. Finally, the dynamic thermomechanical behaviour of the materials is in good agreement with those of mechanical characterization, and this is explained by the compatibilizing effect of the studied resins [49].

### 3.6. Wettability

Figure 5 shows the variation of the PBAT water contact angle with an increasing content of GR and UT (from 5 to 15 wt.%). PBAT has a water contact angle (WCA) of 75.8°, as it possessed hydrophobic characteristics [51]. It is known that WCA higher than 65° is typical in hydrophobic surfaces, while WCA values lower than 65° are obtained in hydrophilic materials [62,63,64]. On the other hand, gum rosin is known to be an amphipathic material, because the resinic acids mixture that composes GR has both hydrophilic and hydrophobic parts [65]. Additionally, gum rosin esters are reported to be more hydrophobic than gum rosin, because the ester linkages present limitations in the formation of hydrogen bonds [48]. Therefore, the addition of either GR or UT significantly uplift (*p* < 0.05) the hydrophobicity of PBAT, as seen in Figure 5, due to the inherent hydrophobicity of gum rosin [37]. GR in 5 and 10 wt.% increased the hydrophobicity of PBAT in 24%, while for PBAT-15GR, the WCA decreased with respect to the previous resin formulations, but being still higher than that of neat PBAT in 16%. This implies that the non-well solubilized GR changes the topography of the PBAT-15GR formulation and the WCA is finally governed by the GR component. Meanwhile, UT in 5 and 15 wt.% further increased the hydrophobicity of PBAT due to the more hydrophobic character with respect to GR, even so to a similar degree of the materials with 5 and 10 wt.% of GR (*p* > 0.05). Regarding the industrial application of the obtained materials, the results point out the high interest for these materials for packaging applications, as low hydrophilicity is desired in materials for this use [7,66].

### 3.7. Colour Characterization

Table 2 summarizes the colour coordinates of PBAT and PBAT resin-based formulations. The colour characterization reveals that GR and UT significantly reduce the lightness, L*, of PBAT (*p* < 0.05) with the increase in the resin content. Additionally, GR reduces the lightness to a greater degree than UT because its natural colouration is stronger. Regarding the a* coefficient, a significant increase is seen as the content of GR increases, pointing out that GR provides a reddish colouration to the material with respect to neat PBAT [67]. On the contrary, UT did not significantly change the a* coefficient of PBAT (*p* > 0.05). The b* coefficient significantly increases with the addition of both GR and UT (*p* < 0.05), which indicates that the material acquires a yellowish hue [67]. Consequently, the yellow index (YI) presents the same behaviour as b*. Again, PBAT-GR formulations present higher values of YI than PBAT-UT formulations due to the colouration differences of both additives. Finally, the total colour differences (∆E) present statistical differences between all the samples (*p* < 0.05) showing values of ∆E higher than 2, which means that the change in colour in the formulations is appreciable to the human eye [37,68]. The variation in the colour of PBAT due to the addition of GR or UT is because the inherent yellowish coloration of the additives [69,70]. Nevertheless, it should be mentioned that although other packaging applications were transparent and colourless formulations are highly desired (i.e., films), for injection moulded rigid packaging applications, the tendency to yellow does not represent a limitation. In fact, the surface colour appearance of the sample specimens is presented in Figure 6 and it can be observed that the formulated materials possess a visual appearance of several commercial rigid packaging applications, particularly PBAT-UT formulations.

## 4. Conclusions

PBAT was blended with two pine resin derivatives, gum rosin (GR) and a pentaerythritol ester of gum rosin (UT), and successfully processed into rigid packaging materials by the injection moulding process. The tensile test results established that PBAT-10GR and PBAT-15UT have tensile strength and toughness at least equal or significantly higher than those of neat PBAT. A plasticizing effect of GR and UT in the PBAT polymeric matrix was observed in all the studied resin contents. The scanning electron micrographs revealed good compatibility between GR and UT with the PBAT polymeric matrix, since no phase separations were observed, except in PBAT-15GR where some signs of insolubilized GR were observed.

Moreover, pine resin derivatives improve the processability of PBAT not only by providing plasticizing effects, but also by reducing the melting enthalpy, which suggests a decrease in the energy consumption when processing theses blends. The thermogravimetric analysis allowed the determination of an enhancement in the thermal stability of PBAT with the addition of either GR (in 5 and 10 wt.%) or UT (in all the studied proportions). DMTA characterization revealed that GR has somewhat better compatibility than UT when blended with PBAT, ascribed to GR’s solubilizing effect into the PBAT matrix, in good agreement with the mechanical results. The colour of neat PBAT shows significant differences with the PBAT–pine resin derivatives blends, due to the intrinsic colouration of the additives, which was less marked in the case of the PBAT-UT blends. The pine resin derivatives increased the hydrophobicity of PBAT in all the developed formulations. In brief, the incorporation of pine resin derivatives (GR and UT) improved the PBAT processability and can reduce the processing and production costs of PBAT formulations by reducing the amount of this material in the final compound, while keeping or even improving its mechanical performance. The hydrophobicity of PBAT was increased, offering a good outlook for injection moulded materials for biodegradable or compostable food packaging applications, where improved surface hydrophobicity is desired.

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
