# Peer review of "Improvement of PBAT Processability and Mechanical Performance by Blending with Pine Resin Derivatives for Injection Moulding Rigid Packaging with Enhanced Hydrophobicity"

_polymers, 2020, doi:10.3390/polym12122891_

Round 1
Reviewer 1 Report
General comments
The submitted manuscript reports on the evaluation of the influence of the addition of gum rosin (GR) and pentaerythritol ester of GR (UT) on the PBAT properties and cost.
The proposed topic well matches the aim and scope of the Polymers, but it cannot be accepted in the current version but needs major revisions.
A deep and accurate English grammar revision, since many mistakes are present.
More details and specific remarks and suggestions are reported below point by point.
Abstract
Keywords
The chosen keywords (i.e. Biopolymers, polybutylene adipate-co-terephthalate (PBAT); pine resin; blends; compatibility effect; plasticizing effect; hydrophobicity). do not completely cover the review content. For example one keyword for the final application has to be added.
Moreover, ‘biopolymers’ can be removed, since too generic.
- Introduction
The Introduction section is too long and redundant. It has to be properly summarised.
- The following statement “Thus, biodegradable polymers have gain considerably attention during the last years, particularly for short term applications, such as food packaging materials” needs suitable references, including “Eco-sustainable systems based on poly(lactic acid), diatomite and coffee grounds extract for food packaging, Intern J Biolog Macromolecules 112, June 2018: 567-575” and “Biodegradable zein film composites reinforced with chitosan nanoparticles and cinnamon essential oil: physical, mechanical, structural and antimicrobial attributes, Colloids and Surfaces B: Biointerfaces 177(2019), 1 May 2019, Pages 25-32.”.
- The originality and added value to the scientific community of the present review has to be better evidenced, at the end of the Introduction section.
- Materials and Methods
2.2.2. Mechanical characterization
- The shape and dimensions of samples for tensile tests have to be specified.
2.2.6. Surface characterization
- The volume of the water drop has to be specified.
- Results
3.2. Microstructural characterization
In Figure 2 the same magnification has to be used for all the compared SEM micrographs.
3.3. Thermal characterization
- The following consideration “This result implies that pine resin derivatives improve the processability of PBAT as less energy is required to melt the blend” has to be corroborated with suitable references.
- Similarly, the following statement “Moreover, PBAT-15GR, present a separation in the melting endothermic peak which is attributed to a disjointing of PBAT and GR fractions” requires proper references.
- In the phrase “The onset degradation temperature is increased in PBAT-10GR and PBAT-UT blends in at least 10 °C”, did the Authors mean all the PBAT-UT compositions with PBAT-UT?
- The conclusion “However, PBAT-15GR has a lower T5% than PBAT which may be due to separation of the components because of the saturation effect of the gum rosin in the PBAT matrix” has to be corroborated with appropriate references.
3.5. Surface characterization
- The following conclusion “PBAT has a water contact angle (WCA) of 75.8 °, as it possessed hydrophobic characteristics [52]” is not correct, since a material is considered hydrophobic when WCA is higher than 90 °.
-The recorded differences due to the different UT and GR amounts have to be better justified.
- The following consideration “Regarding to the industrial application of the obtained materials, the results point out the high interest for this materials for packaging applications, as low hydrophilicity is desired in materials for this use” needs suitable references, including “A comprehensive review on the nanocomposites loaded with chitosan nanoparticles for food packaging, Critical Reviews in Food Science and Nutrition 2020”.
Conclusions
The Conclusions section is too long. It has to be summarised, since in some points it sounds more as a Results and Discussion section, highlighting the main achievements.
Author Response
General comments
The submitted manuscript reports on the evaluation of the influence of the addition of gum rosin (GR) and pentaerythritol ester of GR (UT) on the PBAT properties and cost.
The proposed topic well matches the aim and scope of the Polymers, but it cannot be accepted in the current version but needs major revisions.
A deep and accurate English grammar revision, since many mistakes are present.
More details and specific remarks and suggestions are reported below point by point.
We thank reviewer for his/her valuable comments and for considering our work suitable for its publication in Polymers after the proposed improvements.
Abstract
Keywords
Point 1: The chosen keywords (i.e. Biopolymers, polybutylene adipate-co-terephthalate (PBAT); pine resin; blends; compatibility effect; plasticizing effect; hydrophobicity). do not completely cover the review content. For example one keyword for the final application has to be added. Moreover, ‘biopolymers’ can be removed, since too generic.
Response 1: Thank you for the comment. The keyword biopolymers was removed, and the keyword packaging was included in reference to the final application.
- Introduction
Point 2: The Introduction section is too long and redundant. It has to be properly summarised.
Response 2: Thank you for this observation, the introduction section has been shortened in the current version of the manuscript
Point 3: The following statement “Thus, biodegradable polymers have gain considerably attention during the last years, particularly for short term applications, such as food packaging materials” needs suitable references, including “Eco-sustainable systems based on poly(lactic acid), diatomite and coffee grounds extract for food packaging, Intern J Biolog Macromolecules 112, June 2018: 567-575” and “Biodegradable zein film composites reinforced with chitosan nanoparticles and cinnamon essential oil: physical, mechanical, structural and antimicrobial attributes, Colloids and Surfaces B: Biointerfaces 177(2019), 1 May 2019, Pages 25-32.”.
Response 3: Thank you for the requirement. The suggested references were added to the manuscript.
Point 4: The originality and added value to the scientific community of the present review has to be better evidenced, at the end of the Introduction section.
Response 4: we have now better explain the main novelty of the present research at the end of the introduction section.
- Materials and Methods
2.2.2. Mechanical characterization
Point 5: The shape and dimensions of samples for tensile tests have to be specified.
Response 5: Thank you for the suggestion. The shape and dimensions of the samples are now specified in the manuscript.
2.2.6. Surface characterization
Point 6: The volume of the water drop has to be specified.
Response 6: Thank you for the observation. The volume of the water drop is now specified.
- Results
3.2. Microstructural characterization
Point 7: In Figure 2 the same magnification has to be used for all the compared SEM micrographs.
Response 7: Unfortunately, due to the different height of the cryofractured surfaces and due to the fact that the SEM service of our university does not allow to get closer to prevent microscopy damage, we decide to observe each sample at the magnifications at which we can proper focus them. In fact, we had big problems focusing the samples and most images resulted out of focus. Therefore, we had to select those magnifications in which it was possible to obtain an adequate focused SEM image. We decide to include the SEM images since, even when, they were taken at different lengths and magnifications, they allow to see the surface, the fracture behavior of the blends and the absence of phase separation.
3.3. Thermal characterization
Point 8: The following consideration “This result implies that pine resin derivatives improve the processability of PBAT as less energy is required to melt the blend” has to be corroborated with suitable references.
Response 8: Thank you for the comment. A suitable reference in now in this sentence.
Point 9: Similarly, the following statement “Moreover, PBAT-15GR, present a separation in the melting endothermic peak which is attributed to a disjointing of PBAT and GR fractions” requires proper references.
Response 9: Thank you for the suggestion, infact we have better explain this point for the safe of clarity as well as a proper reference was added to the manuscript.
Point 10: In the phrase “The onset degradation temperature is increased in PBAT-10GR and PBAT-UT blends in at least 10 °C”, did the Authors mean all the PBAT-UT compositions with PBAT-UT?
Response 10: Thank you for the question. Indeed, it means all the PBAT-UT compositions. This was clarified in the current version of manuscript.
Point 11: The conclusion “However, PBAT-15GR has a lower T5% than PBAT which may be due to separation of the components because of the saturation effect of the gum rosin in the PBAT matrix” has to be corroborated with appropriate references.
Response 11: Thank you for the suggestion, this point has been better explained and a proper reference was now added in the manuscript.
3.5. Surface characterization
Point 12: The following conclusion “PBAT has a water contact angle (WCA) of 75.8 °, as it possessed hydrophobic characteristics [52]” is not correct, since a material is considered hydrophobic when WCA is higher than 90 °.
Response 12: Thank you for the comment. However, we have worked with three different references that specify that a water contact angle higher than 65° is typical in hydrophobic surfaces, while θ values lower than 65° are obtained in hydrophilic materials.
The references are listed below:
Point 13: The recorded differences due to the different UT and GR amounts have to be better justified.
Response 13: Thank you for this comment. We have now better explained the differences between PBAT-UT and PBAT-GR formulations along the manuscript.
Point 14: The following consideration “Regarding to the industrial application of the obtained materials, the results point out the high interest for this materials for packaging applications, as low hydrophilicity is desired in materials for this use” needs suitable references, including “A comprehensive review on the nanocomposites loaded with chitosan nanoparticles for food packaging, Critical Reviews in Food Science and Nutrition 2020”.
Response 14: Thank you for the requirement. The proposed reference is now included in the mentioned paragraph.
Conclusions
Point 15: The Conclusions section is too long. It has to be summarised, since in some points it sounds more as a Results and Discussion section, highlighting the main achievements.
Response 15: Thank you for the suggestion. Some changes were done in the manuscript.

Reviewer 2 Report
Please see the attachment.

Author Response
Point 1: Line 25: Thus? Then?
Response 1: Thank you for the suggestion. We have changed the text in the current version of the manuscript.
Point 2: However, the mechanical performance and the high prices of PBAT limit its usage at commercial level. PBAT was blended with 5, 10, and 15 wt. % of the additives by melt-extrusion followed by injection moulding. A 10 wt.% of GR significantly increased the tensile properties of PBAT while a 15 wt.% of UT kept PBAT tensile performance.
Only 10% of PBAT was substituted by GR, it could not significantly reduce the cost of final product. Please reorganize the abstract and introduction. The main purpose of this study is to improve the mechanical property of film as food packaging, not the price. If PBAT was substituted by additives up to 30%, it could indeed reduce the cost.
Response 2: Thank you for the suggestion. We are not working with films, we are developing injected molded samples. We have reorganized the ideas in the manuscript to better explain the main objective of the work. Moreover, although 10 % of gum rosin is not high amount to considerably reduce the price of the material, at industrial level any small amount of the most expensive component in the formulation that can be reduced is greatly appreciated. Thus, we have better explain this concept in the current version of the manuscript.
Point 3: As could be seen from Scheme 1, the composite film became darker. As a packaging film, it should be transparent enough to see through the film for consumers.
Line 408: Finally, the total colour differences 408 (ΔE) present statistical differences between all the samples (p < 0.05), which means that the change in colour in the formulations is appreciable to the human eye.
How could you obtain the conclusion? The authors should analyze the changes in color to be used as a functional food packaging film.
Thank you for your comments. As we already mentioned we have developed injected moulded rigid materials instead of films. Although transparency is required for film intended for food packaging, it is different for injected moulded rigid parts. This is why we did not measure the transparency. Nevertheless, considering that as you correctly mention we have colour changes that are appreciable to human eye, we have introduce some new comments regarding this point and we think that now the colour results discussion have been improved.

Reviewer 3 Report
This paper reports the effect of using pine resin and rosin esters as biobased additives to improve the processability and several properties of PBTA. The work is the third one of a series of similar publications on PVC, PLA, and Mater Bi modified by the same additives and analyzed similarly. The manuscript is interesting and has valuable practical information, however, the manuscript has several issues and inconsistencies that must be clarified to meet the quality level of the Journal. Consequently, the manuscript is not acceptable in the present form and should be resubmitted after major revisions.
Here below my comment.
- Introduction. This section must be shortened, addressing the attention to the latest relevant results on the subject. For the sake of clarity, please rephrase lines 87-89.
- References. The paper has 63 references. Please check the significance of all of them
- Materials. Provide structures of GR and esters
- Mechanical characterization. Provide the shape and dimensions of the specimens.
- FESEM. Which surface of the specimen is observed? Fracture? Please describe
- Contact angle and color should be described separately.
- Results. Line 211. Include Tg values in your analysis
Line 218. Authors stated: PBAT. This indicates that at 10 wt. % of GR is the saturation point, where PBAT achieved a higher value of the tensile strength. This is not necessarily true because the solubility limit could be found near 10 or between 10 and 15 wt.% GR. Additionally, what do you mean by “saturation effect”? Phase separation due to limited compatibility with the matrix? One way to check the potential compatibility between co-components is by comparing the solubility parameters.
- Lines 255-257. The authors said: “The tensile test results prove that 10 wt.% of GR significantly increased the tensile strength and toughness of PBAT. Meanwhile, 15 wt.% UT preserved PBAT tensile properties. Additionally, UT significantly increases the hardness of PBAT in all the contents while GR does so in 5 and 10 wt. % and keeps the PBAT value at 15 wt.%. Mechanical results evidence that GR and UT “. Why? Can you describe the potential interactions between PBTA components and both additives?
- Thermal characterization. Provide Tg of the parent PBAT. Did you determine the Tg of the blends by DSC?
Lines 305-306. Authors stated: Moreover, PBAT-15GR, present a separation in the melting endothermic peak which is attributed to a disjointing of PBAT and GR fractions”. Is GR crystalline? I think the reason for this split in the melting is attributed to the melting of crystals of different quality… please explain
- TGA. Remove Fig. 4. Relevant experimental data are listed in Table 1. The discussion should be reorganized and shortened. The assumption that phase separation between co-components is the main thing responsible for the decreased T5% must be supported
- DMA. Tg is more likely related to Talpha from the loss modulus curve than from tan delta. Could you provide these values? Lines 370. Why is less compatible with the blend with the lower GR? In the discussion of mechanical properties, you stated that 10wt.%GR is the “saturation point”. Besides, FESEM observations did not give any clear evidence of this assumption. Figure 5b. Explain the peaks observed around 60 ºC
- Line 388. Delete Table 2. This table list color parameters. You describe that the contact angle was improved. Why? Provide an explanation supporting your results.
- Color. Why does color change with the addition of the additives?
- Conclusions. This section is similar to the abstract and should be rewritten. Lines 423-429. This part is unclear and should be rewritten: saturation, phase separation, no phase separation, problems in interfacial adhesion. I think the last paragraph summarizes the conclusion.
Author Response
Comments and Suggestions for Authors
This paper reports the effect of using pine resin and rosin esters as biobased additives to improve the processability and several properties of PBTA. The work is the third one of a series of similar publications on PVC, PLA, and Mater Bi modified by the same additives and analyzed similarly. The manuscript is interesting and has valuable practical information, however, the manuscript has several issues and inconsistencies that must be clarified to meet the quality level of the Journal. Consequently, the manuscript is not acceptable in the present form and should be resubmitted after major revisions.
Here below my comment.
We thank Reviewer 3 for his/her valuable comments and for considering our work suitable for its publication in Polymers after the proposed improvements.
Point 1. Introduction. This section must be shortened, addressing the attention to the latest relevant
Response 1: in the current version of the manuscript the introduction section has been shortened and we reduce the amount of references considering the latest relevance.
Point 2. results on the subject. For the sake of clarity, please rephrase lines 87-89.
Response 2: Thank you for the observation. In the current version of the manuscript we have rephrased the sentence and we think that it is now more clear.
Point 3. References. The paper has 63 references. Please check the significance of all of them
Response 3: Since we have shortened the introduction section, many references have been avoided. However, we have introduced some more new references according with Reviewer 1 suggestions.
Point 4. Materials. Provide structures of GR and esters
Response 4: Thank you for the suggestion. The chemical structures were added in Scheme 1.
Point 5. Mechanical characterization. Provide the shape and dimensions of the specimens.
Response 5: Thank you for the suggestion. The shape and dimensions of the samples are now specified in the manuscript.
Point 6. FESEM. Which surface of the specimen is observed? Fracture? Please describe
Response 6: Thank you for the comment. Indeed, the cryofracture surface of the specimens was observed and it is now properly described in the manuscript.
Point 7. Contact angle and color should be described separately.
Response 7: Thank you for the suggestion. Contact angle and color were divided into different sections and described separately. We think that now the results are better described in this separately way.
Point 8. Results. Line 211. Include Tg values in your analysis
Response 8: Thank you for the requirement. Tg values are now included in the manuscript in Table 1 and the results of Tg2 have been related with the tensile test results and properly included in the analysis of the results
Point 9. Line 218. Authors stated: PBAT. This indicates that at 10 wt. % of GR is the saturation point, where PBAT achieved a higher value of the tensile strength. This is not necessarily true because the solubility limit could be found near 10 or between 10 and 15 wt.% GR. Additionally, what do you mean by “saturation effect”? Phase separation due to limited compatibility with the matrix? One way to check the potential compatibility between co-components is by comparing the solubility parameters.
Response 9:Thank you for this observation. Reviewer is totally true, for the safe of clarity the previous mentioned saturation effect was replaced by a more proper explanation and avoided along the manuscript. Moreover, the solubility parameters of each component in the blend have been calculated and added in the current version of the manuscript.
- Lines 255-257. The authors said: “The tensile test results prove that 10 wt.% of GR significantly increased the tensile strength and toughness of PBAT. Meanwhile, 15 wt.% UT preserved PBAT tensile properties. Additionally, UT significantly increases the hardness of PBAT in all the contents while GR does so in 5 and 10 wt. % and keeps the PBAT value at 15 wt.%. Mechanical results evidence that GR and UT “. Why? Can you describe the potential interactions between PBTA components and both additives?
Response 9.1: In the current version of the manuscript we have added some comments regarding the solubility parameter similarities/differences among components. Moreover, as reviewer suggested some comments regarding the potential chemical interactions were added.
- Thermal characterization. Provide Tg of the parent PBAT. Did you determine the Tg of the blends by DSC?
Response 9.2: Thank you for the suggestion. Tg values are now included in Table 1. Tg values of PBAT and the blends were not obtained from DSC but they were obtained from DMA analyses and in the current version of the manuscript are better discussed.
Point 10. Lines 305-306. Authors stated: Moreover, PBAT-15GR, present a separation in the melting endothermic peak which is attributed to a disjointing of PBAT and GR fractions”. Is GR crystalline? I think the reason for this split in the melting is attributed to the melting of crystals of different quality… please explain
Response 10: Thank you for the comment. The Reviewer is totally right, the reason for the double melting peak is due to the formation of different crystal morphologies, because amorphous GR is no longer solubilized in the PBAT matrix and allows less stable crystals formation during processing. In the current version of the manuscript, we have better explain this point and proper references along with the explanation were added to the manuscript.
- TGA. Remove Fig. 4. Relevant experimental data are listed in Table 1.
Response 10.1: Thank you for your comments. In the current version of the manuscript the Figure 4 has been moved to the Supporting Information.
- The discussion should be reorganized and shortened. The assumption that phase separation between co-components is the main thing responsible for the decreased T5% must be supported.
Response 10.2: Reviewer is totally rigth. In the current version of the manuscript we have better explained and discussed this point.
- DMA. Tg is more likely related to Talpha from the loss modulus curve than from tan delta. Could you provide these values? Lines 370. Why is less compatible with the blend with the lower GR? In the discussion of mechanical properties, you stated that 10wt.%GR is the “saturation point”. Besides, FESEM observations did not give any clear evidence of this assumption. Figure 5b. Explain the peaks observed around 60 ºC
Response10.3: Thank you for the suggestion. Changes were implemented in the discussion of DMA results in the current version of the manuscript. Moreover, the peaks observed around 60 °C are properly explained in lines 398 to 409.
- Line 388. Delete Table 2. This table list color parameters. You describe that the contact angle was improved. Why? Provide an explanation supporting your results.
Response10.4: Thank you for the comment. The words “Table 2” were deleted from line 388, and the proper reference was placed instead. Regarding the increase on the water contact angle values it was directly related with the hydrophobic nature of the pine resin, while also the surface topography was also considered.
- Color. Why does color change with the addition of the additives?
Response10.5: Thank you for the question. The change in the color of PBAT is due to the inherent colouration of pine resin derivatives. Some comments have been now added to clarify this point.
- Conclusions. This section is similar to the abstract and should be rewritten. Lines 423-429. This part is unclear and should be rewritten: saturation, phase separation, no phase separation, problems in interfacial adhesion. I think the last paragraph summarizes the conclusion.
Response 10.6: We thank this observation. The conclusion section was rewritten and we think that it was considerably improved. The saturation word was changed by a more proper explanation along the manuscript for the safe of clarity and, thus, it was not used in the conclusion section.

Round 2
Reviewer 1 Report
General comments
The manuscript has been really improved. Two minor revisions have to be applied, as reported below.
- Materials and Methods
2.2. Methods
2.2.1. Miscibility prediction
A reference for the reported equation has to be added.